# Synovial Macrophages Expression of OX40L Is Required for Follicular Helper T Cells Differentiation in the Joint Microenvironment

**DOI:** 10.3390/cells11203326

**Published:** 2022-10-21

**Authors:** Xiaoyu Cai, Meng Zhang, Fujia Ren, Weidong Fei, Xiao Zhang, Yunchun Zhao, Yao Yao, Nengming Lin

**Affiliations:** 1Key Laboratory of Clinical Cancer Pharmacology and Toxicology Research of Zhejiang Province, Department of Clinical Pharmacology, Affiliated Hangzhou First People’s Hospital, Cancer Center, Zhejiang University School of Medicine, Hangzhou 310006, China; 2Department of Pharmacy, Women’s Hospital, Zhejiang University School of Medicine, Hangzhou 310006, China; 3Department of Pharmacy, Hangzhou Women’s Hospital, Hangzhou 310006, China

**Keywords:** follicular helper T cells, OX40L, rheumatoid arthritis, differentiation, joint microenvironment

## Abstract

Signaling via the OX40/OX40L axis plays a key role in CD4^+^ T cell development, and OX40L expression is primarily restricted to antigen-presenting cells (APCs). This study was designed to assess the role of APC-mediated OX40L expression in the context of the development of rheumatoid arthritis (RA)-associated CD4^+^ T cell subsets. For these analyses, clinical samples were harvested from patients with osteoarthritis and RA, with additional analyses performed using OX40^−/−^ mice and mice harboring monocyte/macrophage-specific deletions of OX40L. Together, these analyses revealed tissue-specific roles for OX40/OX40L signaling in RA. Specifically, higher levels of synovial macrophage OX40L expression were associated with the enhanced development of T follicular helper cells in the joint microenvironment, thereby contributing to the pathogenesis of RA. This Tfh differentiation was found to be OX40/OX40L-dependent in this synovial setting. Overall, these results indicate that the expression of OX40L by synovia macrophages is necessary to support Tfh differentiation in the joint tissues, thus offering new insight regarding the etiological basis for RA progression.

## 1. Introduction

Rheumatoid arthritis (RA) is a chronic form of autoimmune inflammatory disease [1]. RA affects an estimated 0.2–1% of the global population, including 0.28–0.41% of individuals in China, and ~80% of affected patients are female [2,3]. RA patients experience bone erosion and persistent inflammation of the joints characterized caused by immune cells and fibroblasts present within the synovial tissues [4]. The progressive erosion of joint tissue and consequent joint bone destruction ultimately cause the incapacitation of individuals with RA [5]. As such, further studies of the inflammatory and immune responses that occur within the synovial microenvironment in RA patients are vital in order to better guide the treatment of this debilitating disease.

The T cell co-stimulatory molecule OX40 and its cognate ligand OX40L have attracted broad research interest as therapeutic targets in T cell-mediated diseases [6]. OX40/OX40L is a key regulator of both innate and adaptive immunity and is capable of regulating both macrophage and T cell function [7]. In RA, the OX40/OX40L pathway plays an important role. In RA models and RA patients, OX40 is involved in the development of RA mediated by T lymphocytes [8]. OX40L mAb administration to type II collagen (CII) immunized DBA/1 mice significantly improved disease severity [9]. T lymphocytes in synovial fluid and synovial tissue from RA patients express OX40, and OX40L is expressed on sub-lining cells in synovial tissue [9]. OX40-Fc fusion protein alleviates PD-1-Fc-exacerbated RA by suppressing the inflammatory response [10]. In addition, OX40 plays a pathogenic role in the development of autoimmune arthritis as an alternative co-stimulator of CD4^+^CD28^−^ T cells, suggesting it as a potential target for immunomodulatory therapy in RA [11]. These data suggest that OX40/OX40L plays a key role in the development of RA and that the OX40/OX40L pathway in T cells may be a potential target for the treatment of RA. 

CD4^+^ T cells are key mediators of RA pathogenesis and important components of the joint microenvironment in affected patients [12]. While many researchers have focused on the specific roles that these CD4^+^ T cells play in RA, how interactions between these cells and synovial macrophages (SMs) contribute to RA progression remains poorly understood. Many different factors control the differentiation of CD4^+^ T cells, including T cell receptor signaling and major histocompatibility complex (MHC)-mediated antigen presentation [13,14]. Tumor necrosis factor (TNF) superfamily proteins are also key regulators of CD4^+^ T cell differentiation [6], with the OX40/OX40L signaling pathway playing a central role in the activation of particular CD4^+^ T cell subsets [15], as confirmed through studies conducted using OX40L^−/−^ or OX40^−/−^ mice [16]. OX40L expression is evident on T cells, innate lymphoid cells, and a range of antigen-presenting cell (APC) types [17]. Efforts to clarify the importance of OX40/OX40L signaling interactions between CD4^+^ T cells and SMs within the joint microenvironment thus have the potential to better clarify the molecular pathogenesis of RA. 

In the context of autoimmune disease, OX40/OX40L signaling can regulate the induction of CD4^+^ T cell responses, including regulatory T cells (Tregs), T follicular helper (Tfh) cells, and type 1 helper T (Th1) cells [18,19,20]. This study was designed to explore the OX40/OX40L signaling that occurs between macrophages and CD4^+^ T cell subsets within the joint microenvironment. Together, these analyses have the potential to offer new insight regarding the molecular etiology of RA-related joint damage.

## 2. Materials and Methods

### 2.1. Ethics Approval

All animal studies were performed in accordance with appropriate ethical guidelines and were approved by the Institutional Animal Care and Use Committee of Zhejiang Laboratory Animal Center (approval no. ZJCLA-IACUC-20040013). Human tissue samples were collected from patients with RA or osteoarthritis (OA) who provided full informed consent to participate. All studies involving patient samples were approved by the Ethics Committee of Women’s Hospital Zhejiang University School of Medicine (approval no. IRB-20200355-R).

### 2.2. Mice

Female DBA/1 mice 5–8 weeks old were purchased from GemPharmatech Co., Ltd. (Nanjing, China). OX40^−/−^ mice were purchased from GemPharmatech Co., Ltd. (Nanjing, China). *Tnfsf4*^fl/fl^ mice and B6/JGpt-*Lyz2^em1Cin(Cre)^*/Gpt mice were purchased from GemPharmatech Co., Ltd. (Nanjing, China). The *Lyz2*-Cre strain uses a promoter specific to myeloid cells (monocytes, mature macrophages, granulocytes) to drive codon-optimized Cre (iCre). This strain specifically expresses Cre protein in myeloid cells and the mice targeted can be used as Cre tool mice for the induction of LoxP recombination in myeloid cells. *Lyz2*-cre mice were bred with conditional knockout model mice to delete the gene fragment between two LoxP. Therefore, mating *Tnfsf4*^fl/fl^ mice and B6/JGpt-*Lyz2^em1Cin(Cre)^*/Gpt mice results in mice with conditional deficiency of OX40L in monocytes/macrophages. 

### 2.3. Human Samples

From September 2019 to August 2021, Clinical samples were obtained from 19 OA patients and 16 RA patients (patients undergoing joint replacement), including 7 and 9 with recurrent- and progressive-type disease, respectively, defined according to the clinical classification criteria of the American Rheumatism Association for knee OA [21,22]. Inclusion criteria for OA: (a) Patients diagnosed with TMJ-OA according to DC/TMD diagnostic criteria; (b) patients should be ≥18 years old at the time of signing the informed consent form, regardless of gender; (c) 18.5 kg/m^2^ ≤ body mass index (BMI) ≤ 35 kg/m^2^, and weight ≥ 50 kg for men and ≥45 kg for women; (d) no TMD-related treatment; (e) patients fully understand the purpose and requirements of the trial, voluntarily participate in the clinical trial, and sign a written informed consent. Exclusion criteria for OA: (a) Unable to walk independently, unable to participate in the study due to dysfunction; (b) knee pain caused by trauma; (c) history of knee surgery; (d) history of rheumatoid arthritis. Inclusion criteria for RA: (a) Age 18 to 65 years, regardless of gender; (b) meet the 2010 American College of Rheumatology (ACR)/European League for Rheumatology (EULAR) classification criteria with an ACR functional classification of I-III; (c) at screening, active RA is defined as at least 6/68 joints with pressure or pain on movement and at least 4/66 joints with swelling; (d) at screening, erythrocyte sedimentation rate (ESR) ≥ upper limit of normal (ULN), or C-reactive protein (CRP) > upper limit of normal (ULN); (e) body mass index [BMI = weight/height squared (kg/m^2^)] within the range of 18–30; (f) fully informed about the study, participate voluntarily, and have signed a written informed consent form. Exclusion criteria for RA: (a) Other types of arthritis (such as primary arthritis, post-traumatic osteoarthritis, gouty osteoarthritis, hemophilic osteoarthritis, and tuberculous arthritis); (b) bilateral knee arthroplasty (RA patients); (c) severe cardiovascular disease (such as myocardial infarction, atrial fibrillation, angina pectoris, and cardiac failure) or cerebrovascular disease (such as cerebral infarction and cerebral hemorrhage); (d) treated with bDMARD within 6 months, prolonged use of oral anticoagulant drugs (such as aspirin, warfarin, and clopidogrel). Participants’ age, smoking, BMI (body mass index), alcohol consumption, common chronic conditions (e.g., diabetes and hypertension), and drug use did not differ significantly between the RA group and osteoarthritis patients group (OA group) (Appendix A).

### 2.4. Induction of Collagen-Induced Arthritis and Animal Samples Collection

A murine collagen-induced arthritis (CIA) model was established as in prior reports [1]. Briefly, mice received primary and secondary immunizations on days 0 and 21, respectively. For mice in the CIA model group, these immunizations consisted of 0.1 mL of complete Freund’s adjuvant (containing 2 mg/mL chicken type II collagen (Chondrex, America, Catalog # 20011) and 4 mg/mL BCG (Hangzhou Prevention Centre, Hangzhou, China)). Beginning on day 28, two researchers independently assessed the arthritic symptoms and body weight of each mouse in this study. Arthritis severity was scored as follows: 0—normal, no joint swelling; 1—mild ankle joint or wrist swelling, or obvious swelling of the fingers; 2—moderate ankle joint or wrist swelling; 3—severe redness and swelling of the whole paw; 4—severe redness and swelling affecting more than 1 joint. Scores for each paw were summed to produce an overall score for each mouse, with a maximum possible score of 16. Mice were euthanized on day 49, at which time the spleens, hind legs, and synovial tissues of these animals were harvested for analysis. 

### 2.5. Cell Sorting

Peripheral blood and single-cell suspensions of synovial tissues from RA patients and OA patients were used for cell sorting. The kit used for the isolation of peripheral blood mononuclear cells (PBMCs) was purchased from Miltenyi (cat. # 130-115-169). Tissue homogenizer (Servicebio, cat. # KZ-II) was used to make single-cell suspensions of synovial tissue for subsequent sorting. To ensure cell activity, we took the following measures: (1) When preparing single cells, we always ensured that the cells were in a 4 °C environment; (2) when resuspending cells, 1–2% fetal bovine serum was added to PBS; (3) when loading samples, the cells were exposed to room temperature for as little time as possible; (4) before sorting, cells were washed with EDTA-containing PBS to remove calcium ions, and DNase I (1 mg/mL) was added to remove adhesions from dead cell DNA; (5) PI staining was used to remove dead cells. Positive sorting was performed by labeling cells with appropriate surface antibodies, followed by flow cytometry-based analyses. The specimen should be fresh and should not contain more than 10% dead cells and debris. Positive sorting steps were as follows. (1) Centrifuge at 300× *g* for 10 min and carefully remove the supernatant as before. (2) Add antibody (10 μL/10^7^ cells) according to instructions and incubate at 4 °C for 15 min in the dark. (3) Add 10–20 times the labeling volume of buffer, wash the cells, centrifuge at 300× *g* for 10 min, remove the supernatant, and repeat the wash 1 time. Add 40 μL of beads per 10^8^ cells, add 960 μL of buffer, and incubate for 15 min at 4 °C, protected from light. Wash by centrifugation at 300× *g*. Remove supernatant and resuspend with buffer (500 μL/10^8^ cells). (4) Perform magnetic sorting with a positive sorting column, using a Midi head and an LS sorting column. The magnetic beads used for this study were from Miltenyi, and included the following: Human CD19 MicroBeads (cat. # 130-050-301) (B cells), Human CD141 MicroBeads (130-090-512) (DCs), Human CD14 MicroBeads (cat. # 130-050-201) (monocytes), Human F4/80 MicroBeads (cat. # 130-110-443) (monocytes).

### 2.6. Flow Cytometry

Flow cytometry was used to analyze sorted cells (Human CD19^+^ B cells, Human CD141^+^ DCs, Human CD14^+^ monocytes, Human F4/80^+^ monocytes) or single-cell suspension of peripheral blood mononuclear cells or single-cell suspension of synovial tissues or single-cell suspension of spleen. Tissue homogenizer (Servicebio, cat. # KZ-II) was used to make single-cell suspensions of spleen. Cells were suspended in 100 μL PBS and stained with appropriate cell surface antibodies (0.5–1.5 μL) for 30 min at 4 °C in the dark. Cells were then rinsed two times using PBS and resuspended in 100 μL of PBS for analysis with a flow cytometer. Antibodies used for this analysis included the following: Hu CD11b APC M1/70 (BD Pharmingen, Cat. # 553312), Hu CD192 BV480 LS132.1D9 (BD Pharmingen, Cat. # 747852), Hu CD11c PE B-ly6 (BD Pharmingen, Cat. # 555392), Hu CD19 FITC HIB19 (BD Pharmingen, Cat. # 555412), Hu CD14 IHC Pure M5E2 (BD Pharmingen, Cat. # 550376), Hu CD183 BV480 (BD Pharmingen, Cat. # 746283), T-BET PE 4B10 (BD Pharmingen, Cat. # 561265) (Suitable for humans and mouse), Hu CD185 Alexa (BD Pharmingen, Cat. # 558113), Hu BATF PE (BD Pharmingen, Cat. # 27120S), Hu CD4 FITC (BD Pharmingen, Cat. # 550628), Hu CD25 APC (BD Pharmingen, Cat. # 560987), Hu Foxp3 PE (BD Pharmingen, Cat. # 560046), Ms CD183 BV750 (BD Pharmingen, Cat. # 747298), Ms CD185 APC (BD Pharmingen, Cat. # 560615), BATF PE (BD Pharmingen, Cat. # 564503), Ms CD25 APC (BD Pharmingen, Cat. # 557192), Ms Foxp3 PE (BD Pharmingen, Cat. # 560408), Ms CD4 FITC (BD Pharmingen, Cat. # 553046).

### 2.7. RT-qPCR

The sorted cells were used for the RT-qPCR analysis. Then, total RNA was extracted from the cells for RT-qPCR experiments. In this study, Trizol was used to extract total RNA. In single-cell suspensions, Trizol preserves RNA integrity while destroying cells and lysing cellular components. After chloroform was added and centrifuged, the lysate was stratified into aqueous and organic phases, with RNA present in the aqueous phase. Total RNA was quantified by a Thermo Scientific NanodROP 2000 spectrophotometer (Thermo Scientific, Waltham, MA, USA). The extracted RNA was then reverse-transcribed using the TAKARA (Japan) reverse transcription kit to synthesize cDNA. The reactions were prepared using SYBRGreen qPCR Master Mix^®^ (TAKARA, Shiga, Japan) in a Pikoreal 96 Realtime PCR System (Thermo Scientific, USA) according to the instructions of the kit to detect the level of mRNA. The first-strand cDNA was synthesized using the Prime Script RT^®^ kit (Takara, Shiga, Japan) according to the instructions. Reverse transcription conditions: 15 min at 37 °C, 5 s at 85 °C, 10 min at 37 °C. RT-qPCR was performed using the one-step method Thermo Step One^®^ at 95 °C for 10 min, followed by 40 cycles at 95 °C for 15 s and 40 cycles at 60 °C, and approximately 1 min at 60 °C. The Bio-Rad PCR platform (T100PCR) was used. The number of technical replicates was 3. CT values of target and internal reference genes were obtained after real-time amplification. GAPDH is used as an internal standard. Primer information of *Tnfsf4* (5’→3’): forward primer GGTCAGGTCTGTCAACTCCTT, reverse primer CATCCAGGGAGGTATTGTCAGT. Relative expression was compared using the ΔΔCT method with appropriate normalization as follows: A = CT(target gene, experimental sample) − CT(internal standard gene, experimental sample), B = CT(target gene, control sample) − CT(internal standard gene, control sample), K = A − B, relative target gene expression = 2 − K. 

### 2.8. Histology

Hind leg and spleen tissue samples harvested from mice were fixed for 24 h in 4% paraformaldehyde. Spleen samples were then immediately paraffinized. Hind limb samples were decalcified for 1 month prior to paraffinization and isolation of the knee and ankle joints. The tissue was placed in a perforated PE tube and then placed in a decalcifying bucket, poured full of EDTA decalcifying solution, sealed, and placed in a constant temperature shaker with a decalcifying solution change cycle of 2–3 days. Paraffinized knee and ankle sections were utilized for H&E staining and safranin O-fast green staining, while spleen sections were utilized for H&E staining. Two researchers independently scored the staining results for these sections. HE staining steps were as follows. (1) Dewaxing of paraffin sections to water: sequentially put the sections into xylene Ⅰ for 20 min, xylene Ⅱ for 20 min, anhydrous ethanol Ⅰ for 5 min, anhydrous ethanol Ⅱ for 5 min, 75% ethanol for 5 min, tap water washing. (2) Hematoxylin staining: stain sections with hematoxylin solution for 3–5 min, rinse with tap water. Then, treat the section with hematoxylin differentiation solution, rinse with tap water. Treat the section with hematoxylin Scott tap bluing, rinse with tap water. (3) Eosin staining: 85% ethanol for 5 min; 95% ethanol for 5 min. Finally stain sections with eosin dye for 5 min. (4) Dehydration and sealing: dehydrate as follows: 100% ethanol I for 5 min, 100% ethanol II for 5 min, 100% ethanol III for 5 min, xylene I for 5 min, xylene II for 5 min, and finally seal with neutral gum. (5) Microscopic examination, image acquisition and analysis. Safranin O-fast green staining steps were as follows. (1) Paraffin sections dewaxed to water: sequentially put the sections into environmentally friendly dewaxing transparent solution Ⅰ for 20 min, environmentally friendly dewaxing transparent solution Ⅱ for 20 min, anhydrous ethanol Ⅰ for 5 min, anhydrous ethanol Ⅱ for 5 min, 75% ethanol for 5 min, and wash with tap water. (2) Fast green staining: section into bone tissue solid green staining solution for 1–5 min, wash with water to remove excess staining solution until the cartilage is colorless, soak in 1% hydrochloric acid ethanol for 10 s, wash slightly with tap water. (3) Saffron staining: the slides were stained in saffron dye solution for 1–5 s, and then put into four cylinders of absolute ethanol, for rapid dehydration for 5 s, 2 s, and 10 s, and kept in the fourth cylinder. (4) Transparent sealing: clean xylene transparent 5 min, neutral resin sealing. (5) Microscopic examination, image acquisition and analysis.

H&E staining scoring criteria for knee and ankle samples were based on the degree of synovial cell hyperplasia (0, no hyperplasia; 1, mild hyperplasia; 2, moderate hyperplasia; 3, severe hyperplasia), the degree of vascular hyperplasia (0, no vascular hyperplasia; 1, mild vascular hyperplasia; 2 moderate vascular hyperplasia; 3 severe vascular hyperplasia), the degree of fibrous tissue hyperplasia (0, no fibrous tissue hyperplasia; 1 mild fibrous tissue hyperplasia; 2, moderate fibrous tissue hyperplasia; 3, severe fibrous tissue hyperplasia), and the degree of lymphocytic infiltration (0, no infiltration; 1 mild infiltration; 2, moderate infiltration; 3 severe infiltration). Safranin O-fast green staining of ankle and knee sections was used to visualize bone tissue, articular cartilage, and subchondral bone structures, with adult bone and cartilage stained green and red, respectively. H&E staining of spleen samples was used to assess the numbers and sizes of germinal centers in the spleen. Scoring was performed by two independent researchers (skilled in scoring joint HE staining), and the results were taken as the mean value.

### 2.9. Statistical Analysis

Data are means ± standard deviation (SD) and were compared using Student’s *t* test (two groups) or one-way ANOVA (three or more groups) after meeting the requisite assumptions for these statistical tests. Numbers of mice in individual experiments are indicated in the figure legends, and all mouse group assignments were random. A blinded approach was used when scoring clinical and histological samples. *p* < 0.05 was the significance threshold, and all analyses were performed using GraphPad Prism (v 8.01) for Windows.

## 3. Results

### 3.1. OX40L Is Up-Regulated in Antigen-Presenting Cells in the Synovial Tissues of RA Patients, and OX40L Level in CD11b^+^CD192^+^ SMs Is Positively Correlated with DAS28

For this study, we collected samples of peripheral blood and synovial tissue from 16 RA patients (seven recurrent type and nine progressive type) and 19 OA patients according to inclusion and exclusion criteria. Initially, we analyzed OX40L expression in synovial APCs from these samples by using positive bead-based sorting to isolate single-cell suspensions of F4/80^+^ SMs, CD19^+^ B cells, and CD141^+^ dendritic cells (DCs). Relative to corresponding cell populations from patients with OA, OX40L expression was significantly increased in synovial tissue F4/80^+^ SMs (Figure 1A), CD19^+^ B cells (Figure 1B), and CD141^+^ DCs (Figure 1C) from RA patients. Flow cytometry also revealed a higher frequency of CD11b^+^CD192^+^ SMs (Figure 1D), CD19^+^ B cells (Figure 1E), and CD11b^+^CD11c^+^ DCs (Figure 1F) in synovial tissue samples from individuals with RA relative to individuals with OA. To extend these analyses further, we sorted monocytes (CD14^+^), B cells (CD19^+^), and DCs (CD141^+^) from patient PBMCs. As in synovial tissues, significantly a higher level of OX40L expression was detected in the monocytes (Figure 1G), DCs (Figure 1H), and B cells (Figure 1I) of RA patients relative to OA patients, and RA patients also exhibited a higher frequency of circulating CD11c^+^ DCs (Figure 1J), CD19^+^ B cells (Figure 1K), and CD14^+^ monocytes (Figure 1L) as compared to OA patients. Notably, OX40L in SMs only showed an increase in synovial tissues of RA patients compared to peripheral blood (Figure 1O). Furthermore, the relationship between OX40L level and DAS28 scores (Figure 1M) was analyzed. Interestingly, the OX40L level in CD11b^+^CD192^+^ SMs in synovial tissues was positively correlated with DAS28 (Figure 1N), whereas other cells were not (Appendix A). These data may suggest that OX40L plays a tissue-specific role in the synovium in individuals with RA, shaping optimal T cell activation in this compartment. These results also support a relationship between OX40L expression and joint damage severity in RA patients.

### 3.2. OX40L Level in SMs Is Positively Correlated with the Proportion of Tfh in CD4^+^ T Cells in the Synovial Tissues of the Patients with RA

To better understand the association between the expression of OX40L by SMs and RA-related disease progression, we next analyzed three different CD4^+^ T cell subsets that are closely linked to OX40/OX40L signaling activity (Th1, Tregs, and Tfh) [19,23,24], using a flow cytometry-based approach to assess the relative frequencies of these three different subsets as a percentage of the overall CD4^+^ T cell population in RA patient synovial tissue samples. Relative to synovial tissue samples from OA patients, those from RA patients contained increased percentages of CD183^+^T-bet^+^ Th1 (Figure 2A) and CD185^+^BATF^+^ Tfh (Figure 3B) cells, whereas the frequency of CD25^+^Foxp3^+^ Tregs (Figure 2C) was reduced. Notably, a positive correlation was observed between OX40L level in SMs and CD185^+^BATF^+^ Tfh frequency (Figure 2D), whereas no such correlation was observed for CD183^+^T-bet^+^ Th1 (Figure 2E) or CD25^+^Foxp3^+^ Tregs (Figure 2F) in RA patients. These data indicate that increased OX40L expression by SMs may play a role in promoting Tfh differentiation in a manner that ultimately promotes increased RA disease activity.

### 3.3. Tfh Differentiation Requires OX40/OX40L Signaling in the Joint Microenvironment

Many different factors can shape CD4^+^ T cell activation and differentiation, including both TCR and OX40/OX40L signaling [24,25,26]. To further explore the importance of OX40/OX40L signaling in the pathogenesis of RA, we established a collagen-induced arthritis (CIA) model by using OX40^−/−^ mice. Strikingly, OX40^−/−^ mice exhibited significantly decreased arthritic disease severity as compared to wild-type (WT) animals (Figure 3A). This coincided with the alleviation of knee pathology in these animals, with lower levels of bone damage, synovial hyperplasia, and inflammatory cell infiltration (Figure 3B,C). Specifically, the wild-type mice had severe cartilage damage, marked synovial hyperplasia, and multiple inflammatory cell infiltrates in the joint cavity, whereas the OX40^−/−^ mice had less cartilage damage, less marked synovial hyperplasia, and a few inflammatory cell infiltrates in the joint cavity. Synovial tissue single-cell suspensions from OX40^−/−^ mice revealed slight decreases in the frequency of CD183^+^T-bet^+^ Th1 as a fraction of total CD4^+^ T cells in the synovial tissues relative to WT mice (Appendix A), with a corresponding increase in CD25+Foxp3+ Treg frequency (Appendix A). Notably, no significant differences in CD183^+^T-bet^+^ Th1 (Appendix A) or CD25^+^Foxp3^+^ Treg (Appendix A) frequencies were observed in the peripheral blood relative to the synovial tissues (Figure 3D). Compared with wild-type mice, the frequency of CD4^+^Tfh in the synovium of OX40^−/−^ mice decreased by 84.8 ± 1.5% (Figure 3E), while the frequency of CD4^+^ Tfh in the peripheral blood decreased by 30.8 ± 1.1% (Figure 3F). The percentage of decrease in the frequency of CD4^+^ Tfh in the synovium of OX40^−/−^ mice was significantly different from that in the peripheral blood (Figure 3G). These results align well with findings from RA patients detailed above, suggesting that OX40/OX40L signaling plays a tissue-specific role in RA, supporting Tfh differentiation within the joint microenvironment in arthritic model mice.

### 3.4. Tfh Differentiation Is Dependent on OX40L Expression by SMs in the Joint Microenvironment

As OX40L expression by SMs was positively correlated with DAS28 values and with the frequency of Tfh as a fraction of the overall CD4^+^ T cell population, the SM OX40L expression level may serve as a key driver of Tfh differentiation in the RA-associated joint microenvironment. To test this possibility, we employed mice exhibiting macrophage-specific OX40L deletion (*Tnfsf4*^fl/fl^/*Lyz2*-Cre mice) and control *Tnfsf4*^fl/fl^ mice to establish a CIA model as above. After 49 days, these mice were euthanized for downstream analyses. *Tnfsf4*^fl/fl^/*Lyz2*-Cre mice exhibited significant reductions in paw swelling (Figure 4A), inflammatory cell infiltration (Figure 4B), and bone damage (Figure 4C) in the knee joint. Specifically, the *Tnfsf4*^fl/fl^ mice had severe cartilage damage, marked synovial hyperplasia, and multiple inflammatory cell infiltrates in the joint cavity, whereas the *Tnfsf4*^fl/fl^/*Lyz2*-Cre mice had less cartilage damage, less marked synovial hyperplasia, and a few inflammatory cell infiltrates in the joint cavity. Compared to *Tnfsf4*^fl/fl^ mice, the frequency of CD4^+^ Tfh was down-regulated in the knee cavity of *Tnfsf4*^fl/fl^/*Lyz2*-Cre mice (Figure 4D), whereas no corresponding differences in the frequencies of CD183^+^T-bet^+^ Th1 (Figure 4E) or CD25^+^Foxp3^+^ Tregs (Figure 4F) were observed. These findings are consistent with a model wherein the expression of OX40L by SMs within the knee microenvironment is required for Tfh differentiation in this mouse CIA model system, whereas this OX40L signaling is dispensable for Th1 or Treg differentiation.

### 3.5. Tfh, Th1, and Treg Differentiation Is not Dependent on OX40L Expression by Peripheral Blood Monocytes

To further explore the tissue-specific nature of the impact of OX40L expression on SMs in the context of Tfh differentiation, we analyzed PBMCs from these experimental mice. No differences in the frequencies of CD185^+^BATF^+^ Tfh were observed as a fraction of total CD4^+^ T cells in *Tnfsf4*^fl/fl^/*Lyz2*-Cre mice compared to *Tnfsf4*^fl/fl^ mice (Figure 5A). Similarly, there were no significant differences in the proportions of CD183^+^T-bet^+^ Th1 (Figure 5B) or CD25^+^Foxp3^+^ Tregs (Figure 5C) as a fraction of total CD4^+^ T cells. These results suggest that the expression of OX40L on monocytes is dispensable for the differentiation of Tfh in the peripheral blood. While other factors can evidently support Tfh development in the periphery, these same signals fail to facilitate such differentiation in the joint microenvironment in the absence of OX40L. 

### 3.6. Tfh, Th1, and Treg Differentiation Is not Dependent on OX40L Expression by Splenic Macrophages

We additionally analyzed CD4^+^ T cell and CD19^+^ B cell subsets in the spleens of these mice. Relative to *Tnfsf4*^fl/fl^ mice, no significant differences were observed in *Tnfsf4*^fl/fl^/*Lyz2*-Cre mice with respect to germinal center size (Figure 6A), CD19^+^CD23^+^CD24^+^ transitional B cell frequency (Figure 6B), or follicular B cell (CD19^+^CD23^+^CD38^+^) frequency (Figure 6C). Consistently, there was no difference in the proportion of CD185^+^BATF^+^ Tfh (Figure 6D), CD183^+^T-bet^+^ Th1 (Figure 6E), or CD25^+^Foxp3^+^ Tregs (Figure 6F) among CD4^+^ T cells in the spleen of *Tnfsf4*^fl/fl^/*Lyz2*-Cre mice. These findings indicate that Tfh, Th1, and Treg differentiation does not require OX40L expression by splenic macrophages, supporting the tissue-specific importance of monocyte OX40L expression for Tfh differentiation.

## 4. Discussion

Synovial macrophages are essential mediators of the inflammatory activity and bone damage observed in the joints of RA patients [27]. These SM populations are highly heterogeneous such that researchers have defined particular SM subtypes within the synovium [28]. The present results suggest that the expression of OX40L by SMs is vital to the effective development of Tfh within the joint microenvironment in the context of RA. 

The initial analyses in this study were centered around synovial tissue samples from individuals with RA and OA based on the rationale that all medically focused research should be of clinical origin and clinically relevant where possible. OA patients served as controls in this study owing to the challenges associated with collecting synovial tissue samples from individuals without any form of joint disease. OX40L expression on the three most abundant APC populations (DCs, B cells, and monocytes/macrophages) was then analyzed in both the synovium and peripheral blood of these patients. In line with prior work, all three of these cell populations were found to exhibit increased OX40L expression in both the peripheral blood and synovium relative to OA patients. Strikingly, only OX40L expression by SMs was positively correlated with RA patient disease activity scores. These results suggest the possibility that OX40L may play a more localized role in the pathogenesis of RA, rather than shaping this disease at the systemic level. The OX40L-mediated regulation of CD4^+^ T cell activation may thus be a key pathway through which OX40/OX40L signaling can control RA development.

To gain further insight into the observed correlation between OX40L expression by SMs and DAS28 scores, the three CD4^+^ T cell subsets most closely related to OX40/OX40L signaling (Tfh, Th1, and Tregs) were examined, given that they can control RA-related inflammatory activity and immune responses [6,8,19,29]. These analyses revealed a positive correlation between the frequency of CD185^+^BATF^+^ Tfh as a fraction of total CD4^+^ T cells and SM OX40L expression, whereas the same correlative relationship was not detected in peripheral blood samples. This suggested the ability of OX40L expressed by SMs to control inflammation and immune response activation within the joint microenvironment through the regulation of Tfh differentiation in a tissue-specific manner. This model was further supported by the use of OX40^−/−^ mice, which revealed that while the impact of losing OX40 expression was systemic in a CIA model system, the corresponding impact on Tfh differentiation was tissue-specific. Specifically, the differentiation of Tfh was found to be OX40/OX40L signaling-dependent. When this signaling was no longer available, compensatory mechanisms were sufficient to sustain Tfh differentiation in the peripheral blood but not within the joint microenvironment. 

To further explore the importance of OX40L expression by SMs in the context of Tfh differentiation, macrophage-specific deletion of OX40L in mice (*Tnfsf4*^fl/fl^/*Lyz2*-Cre mice) and control mice (*Tnfsf4*^fl/fl^ mice) was used in the study. *Tnfsf4* is the gene encoding the OX40L protein, and *Lyz2* is predominantly expressed in macrophages. *Lyz2*-Cre mice were mated with *Tnfsf4*^fl/fl^ mice to obtain *Tnfsf4*^fl/fl^/*Lyz2*-Cre mice. Our data revealed that differentiation of Tfh requires SMs to express OX40L in the articular microenvironment. Through gene editing studies in mice, our data reveal that the differentiation of Tfh requires SMs to express OX40L in the joint microenvironment. However, this phenomenon was not observed in the peripheral blood and spleen of *Tnfsf4*^fl/fl^/*Lyz2*-Cre mice, which laterally corroborates that the differentiation of Tfh is tissue-specific with respect to the requirement for OX40L.

Tfh are essential mediators of B cell activation and differentiation, promoting germinal center formation and immunoglobulin class switching such that they are closely related to the development of humoral immune responses [30,31]. Tfh differentiation occurs through pathways distinct from those employed by other related CD4^+^ T cell populations such as Th1, Th17, and Tregs [32,33]. The process of Tfh differentiation entails initiation, maintenance, and polarization phases that rely on the coordinated activity of many surface molecules, cytokines, and transcriptional regulators [26]. During the initiation phase, Tfh must interact with APCs. In line with these prior reports, the present data emphasize the fact that Tfh differentiation is distinct from that of Th1 and Tregs.

There are certain limitations to these analyses. For one, only specific APC subsets and CD4^+^ T cell subpopulations were analyzed, potentially resulting in phenotypes associated with other similar cell types having been overlooked. As such, further research will be essential to fully clarify the tissue-specific roles of OX40/OX40L signaling.

## 5. Conclusions

In summary, these results indicate that OX40L is upregulated in APCs isolated from RA patient synovial tissues, and that the expression of OX40L by CD11b^+^CD192^+^ SMs is positively correlated with RA disease severity. The OX40L level in SMs is also positively associated with Tfh frequency in RA patient synovial tissues, while in CIA model mice, the differentiation of Tfh within the joint microenvironment is dependent on OX40/OX40L signaling. In contrast, Tfh, Th1, and Treg differentiation in the peripheral blood of CIA model mice is not dependent on OX40L expression by SMs or splenic OX40L expression. Overall, these data highlight the importance of OX40L expression by SMs as an essential mediator of Tfh development within the RA-associated joint microenvironment.

## Figures and Tables

**Figure 1 cells-11-03326-f001:**
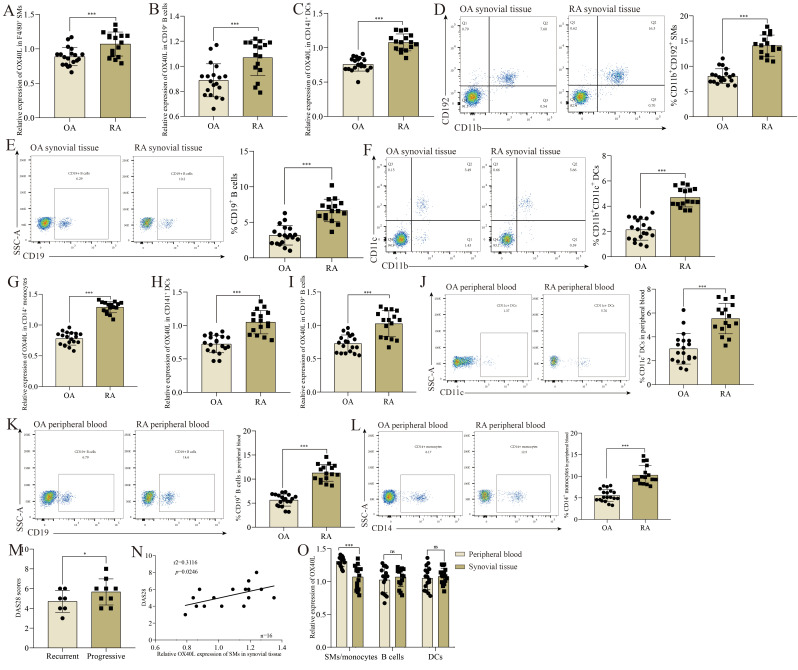
OX40L is up-regulated in antigen-presenting cells in the synovial tissues of RA patients, and the OX40L level in CD11b^+^CD192^+^ SMs is positively correlated with DAS28. (**A**) RT-qPCR analysis of the relative expression of OX40L in F4/80^+^ SMs sorted from synovial tissue. (**B**) RT-qPCR analysis of the relative expression of OX40L in CD19^+^ B cells sorted from synovial tissue. (**C**) RT-qPCR analysis of the relative expression of OX40L in CD141^+^ DCs sorted from synovial tissue. (**D**) Frequency of CD11b^+^CD192^+^ SMs in the synovial tissues of the patients with RA and the patients with OA. (**E**) Frequency of CD19^+^ B cells in the synovial tissues of the patients with RA and the patients with OA. (**F**) Frequency of CD11b^+^CD11c^+^ DCs in the synovial tissues of the patients with RA and the patients with OA. (**G**) RT-qPCR analysis of the relative expression of OX40L in CD14^+^ monocytes sorted from peripheral blood. (**H**) RT-qPCR analysis of the relative expression of OX40L in CD141^+^ DCs sorted from peripheral blood. (**I**) RT-qPCR analysis of the relative expression of OX40L in CD19^+^ B cells sorted from peripheral blood. (**J**) Frequency of CD11c^+^ DCs in the peripheral blood of RA patients and OA patients. (**K**) Frequency of CD19^+^ B cells in the peripheral blood of the patients with RA and the patients with OA. (**L**) Frequency of CD14^+^ monocytes in CD14^+^ monocytes in the peripheral blood of the patients with RA and the patients with OA. (**M**) DAS28 score for RA patients. (**N**) Correlation analysis of OX40L level of SMs in the synovial tissues with DAS28 in RA patients. (**O**) Comparison of OX40L level in the three cell types (SMs/monocytes, B cells, and DCs) in the synovial tissues and peripheral blood. In (**A**–**L**), *** *p* < 0.001 represent a significant difference between the RA group and the OA group. In (**M**), * *p* < 0.05 represents a significant difference between the progressive group and the recurrent group. In (**N**), *p* < 0.05 represents that the vertical coordinate is correlated with the horizontal coordinate. In (**O**), *** *p* < 0.001 represents a significant difference between the peripheral blood group and the synovial tissues group, and ns means no significant difference. Data represent mean ± SD (unpaired *t* test was used).

**Figure 2 cells-11-03326-f002:**
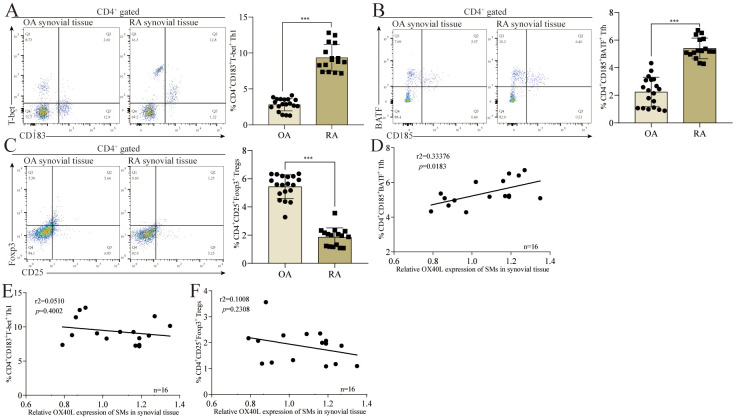
OX40L level in SMs is positively correlated with the proportion of Tfh in CD4^+^ T cells in the synovial tissues of RA patients. (**A**) The proportion of CD183^+^T-bet^+^ Th1 in CD4^+^ T cells in the synovial tissues of RA patients. (**B**) The proportion of CD185^+^BATF^+^ Tfh in CD4^+^ T cells in the synovial tissues of RA patients. (**C**) The proportion of CD25^+^Foxp3^+^ Tregs in CD4^+^ T cells in the synovial tissues of RA patients. (**D**) Correlation analysis of OX40L level in SMs with CD4^+^CD185^+^BATF^+^ Tfh. (**E**) Correlation analysis of OX40L level in SMs with CD4^+^CD183^+^T-bet^+^ Th1. (**F**) Correlation analysis of OX40L level in SMs with CD4^+^CD25^+^Foxp3^+^ Tregs. In (**A**–**C**), *** *p* < 0.001 represents a significant difference between the OA group and the RA group. Data represent mean ± SD (unpaired *t* test was used). In (**D**–**F**), *p* < 0.05 represents that the vertical coordinate is correlated with the horizontal coordinate; otherwise, there is no correlation (linear regression was used).

**Figure 3 cells-11-03326-f003:**
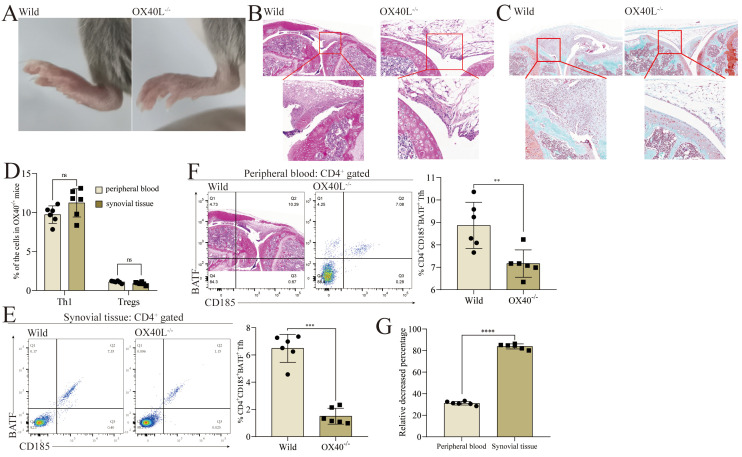
Tfh differentiation is dependent on the OX40/OX40L signaling in the joint microenvironment in mice. (**A**) Collagen-induced arthritis model was established on wild-type mice and OX40L^−/−^ mice. All mice were sacrificed on day 42. The picture shows the paw of the mouse on day 42. (**B**) HE staining of the knee in mice on day 42. (**C**) SafraninO-fast green staining of the knee in mice on day 42. (**D**) The proportion of CD183^+^T-bet^+^ Th1 and CD25^+^Foxp3^+^ Tregs in CD4^+^ T cells in the peripheral blood and the synovial tissues of OX40L^−/−^ mice. (**E**) The proportion of CD185^+^BATF^+^ Tfh in CD4^+^ T cells in the synovial tissues of wild-type and OX40L^−/−^ mice. Left, scatterplot; right, histogram. (**F**) The proportion of CD185^+^BATF^+^ Tfh in CD4^+^ T cells in the peripheral blood of wild-type and OX40L^−/−^ mice. Left, scatterplot; right, histogram. (**G**) Comparison of the proportion of CD185^+^BATF^+^ Tfh in CD4^+^ T cells in the peripheral blood and the synovial tissues. In (**E**,**F**), ** *p* < 0.01 and *** *p* < 0.001 represent a significant difference between the wild-type group and the OX40L^−/−^ group. In (**D**,**G**), **** *p* < 0.0001 represents a significant difference between the peripheral blood group and the synovial tissue group, and ns means no significant difference. Data represent mean ± SD (unpaired *t* test was used).

**Figure 4 cells-11-03326-f004:**
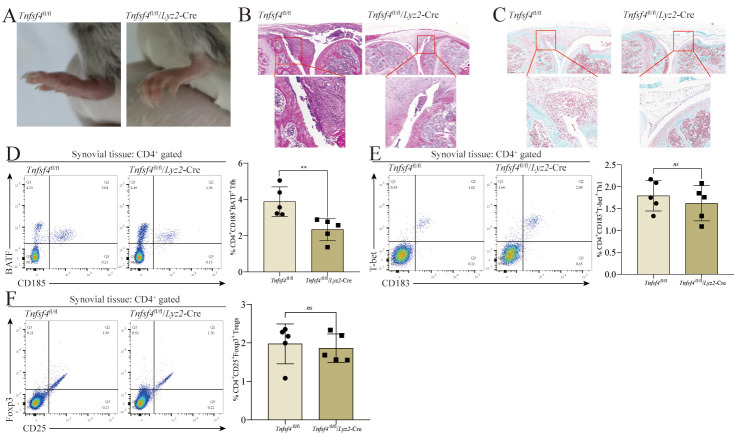
The differentiation of Tfh requires SMs to express OX40L in the joint microenvironment of CIA mice, while Th1 and Tregs do not. (**A**) Photographs of the paw of *Tnfsf4*^fl/fl^/*Lyz2*-Cre mice and *Tnfsf4*^fl/fl^ mice on day 49. (**B**) H&E staining of the knee of *Tnfsf4*^fl/fl^/*Lyz2*-Cre mice and *Tnfsf4*^fl/fl^ mice on day 49. (**C**) SafraninO-fast green staining of the knee of *Tnfsf4*^fl/fl^/*Lyz2*-Cre mice and *Tnfsf4*^fl/fl^ mice on day 49. (**D**) The proportion of CD185^+^BATF^+^ Tfh in CD4^+^ T cells in the synovial tissues. (**E**) The proportion of CD183^+^T-bet^+^ Th1 in CD4^+^ T cells in the synovial tissues. (**F**) The proportion of CD25^+^Foxp3^+^ Tregs in CD4^+^ T cells in the synovial tissues. ** *p* < 0.01 represents a significant difference between the *Tnfsf4*^fl/fl^/*Lyz2*-Cre group and the *Tnfsf4*^fl/fl^ group, and ns means no significant difference. Data represent mean ± SD (unpaired *t* test was used).

**Figure 5 cells-11-03326-f005:**
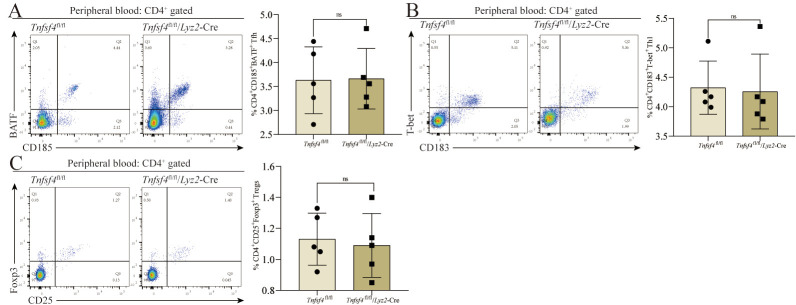
The differentiation of Tfh, Th1, and Tregs does not require SMs to express OX40L in the peripheral blood of CIA mice. (**A**) The proportion of CD185^+^BATF^+^ Tfh in CD4^+^ T cells in the peripheral blood. (**B**) The proportion of CD183^+^T-bet^+^ Th1 in CD4^+^ T cells in the peripheral blood. (**C**) The proportion of CD25^+^Foxp3^+^ Tregs in CD4^+^ T cells in the peripheral blood. ns means no significant difference between the *Tnfsf4*^fl/fl^/*Lyz2*-Cre group and the *Tnfsf4*^fl/fl^ group. Data represent mean ± SD (unpaired *t* test was used).

**Figure 6 cells-11-03326-f006:**
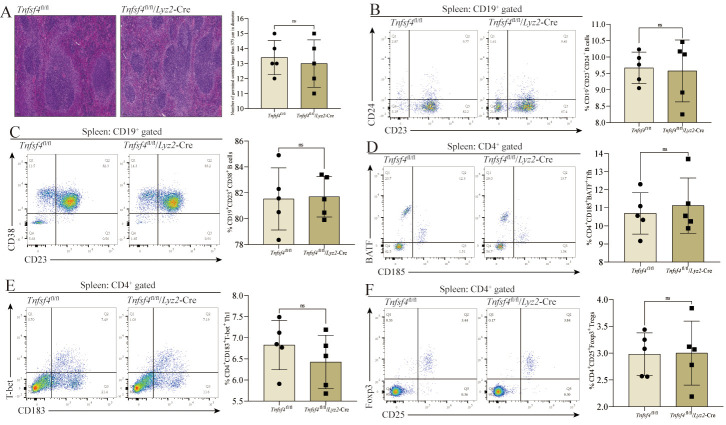
The differentiation of Tfh, Th1, and Tregs does not require macrophages to express OX40L in the CIA mice spleen. (**A**) HE staining of the spleen, and assessment of the size and number of the germinal center. (**B**) The frequency of CD19^+^CD23^+^CD24^+^ transitional B cells in the spleen. (**C**) The frequency of CD19^+^CD23^+^CD38^+^ follicular B cells in the spleen. (**D**) The proportion of CD185^+^BATF^+^ Tfh in CD4^+^ T cells in the spleen. (**E**) The proportion of CD183^+^T-bet^+^ Th1 in CD4^+^ T cells in the spleen. (**F**) The proportion of CD25^+^Foxp3^+^ Tregs in CD4^+^ T cells in the spleen. ns means no significant difference between the *Tnfsf4*^fl/fl^/Lyz2-Cre group and the *Tnfsf4*^fl/fl^ group. Data represent mean ± SD (unpaired *t* test was used).

## Data Availability

The data that support the findings of this study are available from the corresponding author upon reasonable request. Some data may not be made available because of privacy or ethical restrictions.

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
