# Peer review of "Synovial Macrophages Expression of OX40L Is Required for Follicular Helper T Cells Differentiation in the Joint Microenvironment"

_cells, 2022, doi:10.3390/cells11203326_

Round 1

Reviewer 1 Report (New Reviewer)

Herein, Xiaoyu Cai et al. demonstrated the association between OX40L expression on the macrophages and the differentiation of follicular helper T cells in arthritis patients. The authors explored the influences of OX40L expression in the synovial tissues and APCs in the joint microenvironment. They concluded that a higher expression of OX40L would contribute to the differentiation of follicular helper T cells as well as the pathology progression of RA. Based on the content and presentation of the manuscript, the reviewer has several comments listed below.

1. In Figure 1E, the “CD11b” label under the flow cytometry result was misplaced. It should be placed in Figure 1D instead.

2. Figures 1 and 2 exhibited the upregulated OX40L expression in the synovial tissues and APCs of arthritis patients. Due to the high similarity of contents in Figure 1 and Figure 2, the authors should consider binding these two figures into one figure.

3. In the demonstration of figures 4B, 4C, 5B, and 5C, the authors need further word description and comparison of the bone damages, hyperplasia, and cell infiltration in the OX40L -/- mice and wt mice, other than just showing the staining results. Only stating the pathology was alleviated is unconvincing and insufficient.

4. The language of the manuscript needs further editing. Authors extensively used passive voice when describing the results, in which authors should use active voice. Also, we noticed some irregular sentences. For example, in line 306, “To more fully examine the importance of…”.

Based on the opinions, the reviewer believes this work needs minor revisions before the journal accepts the manuscript. 

Author Response

Response to reviewer 1

Herein, Xiaoyu Cai et al. demonstrated the association between OX40L expression on the macrophages and the differentiation of follicular helper T cells in arthritis patients. The authors explored the influences of OX40L expression in the synovial tissues and APCs in the joint microenvironment. They concluded that a higher expression of OX40L would contribute to the differentiation of follicular helper T cells as well as the pathology progression of RA. Based on the content and presentation of the manuscript, the reviewer has several comments listed below.

Response: Thank you for your review and comments.

  1. In Figure 1E, the “CD11b” label under the flow cytometry result was misplaced. It should be placed in Figure 1D instead.

Response: Indeed, we have revised it. Figure 1D.

  1. Figures 1 and 2 exhibited the upregulated OX40L expression in the synovial tissues and APCs of arthritis patients. Due to the high similarity of contents in Figure 1 and Figure 2, the authors should consider binding these two figures into one figure.

Response: We have bonded Figures 1 and 2. Figure 1. Line 300-313.

  1. In the demonstration of figures 4B, 4C, 5B, and 5C, the authors need further word description and comparison of the bone damages, hyperplasia, and cell infiltration in the OX40L -/- mice and wt mice, other than just showing the staining results. Only stating the pathology was alleviated is unconvincing and insufficient.

Response: We have added further word description and comparison in the text in the appropriate place. Line 348-351, 387-391.

  1. The language of the manuscript needs further editing. Authors extensively used passive voice when describing the results, in which authors should use active voice. Also, we noticed some irregular sentences. For example, in line 306, “To more fully examine the importance of…”.

Response: We have changed the passive voice in the results section to the active voice and revised some irregular sentences. And this manuscript has been touched up by a language editing company. Line 267-270, 277-278, 317-318, 342-344, 383-385, 411-412, 429-430.

Based on the opinions, the reviewer believes this work needs minor revisions before the journal accepts the manuscript.

Reviewer 2 Report (New Reviewer)

In this paper the authors demonstrate the role of OX40L on synovial macrophages is important in rheumatoid arthritis. They show synovial macrophage upregulation of OX40L that are important cells in RA pathogenesis.  They further show that it is higher in the peripheral blood monocyte population in RA too. They then show a positive correlation between OX40L synovial macrophages and t follicular helper cells.  they show also in a collagen induced arthritis preclinical model that OX40 KO mice are protected from CIA in this specific model in coincidence with reduce T follicular helper cells suggesting that OX40 pathway is involved in pathogenesis through differentiation of T follicular helper cells. To prove this they used myeloid specific cell deletion of OX40L to delete this specifically in myeloid orgin cells such as macrophages or mast cells.  Here the cell specific Knock outs were protected from CIA compared to the flox controls. In association with the reduced pathology was reduced T follicular helper cells in the synovial tissue by flow cytometry but not Th1 or T regulatory cells. Finally they show this is not a requirement for differentiation of T f cells in the peripheral blood demonstrating a tissue specific role of this.

This is interesting and I have a few comments. 

Does synovial levels of TNF alpha lead to the upregulation of OX40L on synovial macrophages?  Ie addd TNF alpha and at specific time points remove the macrophages and look at the expression of OX40L on the macrophages. 

I would also add a bit more general information in the introduction about the OX40L in general and not specific to RA in the introduction section. Be quite broad as this is not clear in the intro about its broad role in immunity.

How does the synovial macrophages OX40L and OX40 lead to increased differentiation of the T follicular helper cells?  Is it mediated by cytokines? Or is it mediated by changes in epigenetic marks like histone modifications in the genes?

Author Response

Response to reviewer 2

In this paper the authors demonstrate the role of OX40L on synovial macrophages is important in rheumatoid arthritis. They show synovial macrophage upregulation of OX40L that are important cells in RA pathogenesis.  They further show that it is higher in the peripheral blood monocyte population in RA too. They then show a positive correlation between OX40L synovial macrophages and t follicular helper cells.  they show also in a collagen induced arthritis preclinical model that OX40 KO mice are protected from CIA in this specific model in coincidence with reduce T follicular helper cells suggesting that OX40 pathway is involved in pathogenesis through differentiation of T follicular helper cells. To prove this they used myeloid specific cell deletion of OX40L to delete this specifically in myeloid orgin cells such as macrophages or mast cells.  Here the cell specific Knock outs were protected from CIA compared to the flox controls. In association with the reduced pathology was reduced T follicular helper cells in the synovial tissue by flow cytometry but not Th1 or T regulatory cells. Finally they show this is not a requirement for differentiation of T f cells in the peripheral blood demonstrating a tissue specific role of this.

Response: Thank you for your review and comments.

This is interesting and I have a few comments.

Does synovial levels of TNF alpha lead to the upregulation of OX40L on synovial macrophages?  Ie addd TNF alpha and at specific time points remove the macrophages and look at the expression of OX40L on the macrophages.

Response: OX40L belongs to the ligand of tumor necrosis factor superfamily receptor. Previous studies showed that the expression of OX40L in osteoarthritic chondrocytes increased after being stimulated by TNF-α (PMID: 33147588). In the joint microenvironment of RA, TNF-α is a typical proinflammatory cytokine. The stimulation of TNF-α promote the differentiation of macrophages into M1 type (PMID: 32419317). We speculate that in the joint microenvironment of RA, adding TNF-α can help synovial macrophages secrete OX40L to promote inflammation. However, no relevant experiments were conducted in the present study. Further study on the relationship among TNF-α, synovial macrophages and OX40L is of great significance to reveal the role of OX40L in RA.

I would also add a bit more general information in the introduction about the OX40L in general and not specific to RA in the introduction section. Be quite broad as this is not clear in the intro about its broad role in immunity.

Response: We have added some general information about its broad role in immunity to introduction section. Line 41-44.

How does the synovial macrophages OX40L and OX40 lead to increased differentiation of the T follicular helper cells?  Is it mediated by cytokines? Or is it mediated by changes in epigenetic marks like histone modifications in the genes?

Response: The OX40/OX40L is a co-stimulatory molecule for T cells and OX40 (receptor) is highly expressed on the cell membrane of Tfh. In the articular cavity, increased secretion of OX40L (ligand) by synovial macrophages further interacts with OX40 on Tfh to promote the differentiation. In the present study, we have only demonstrated the existence of this phenomenon, while the mechanisms behind this phenomenon have not been explored. In the articular cavity, the specific promotion of Tfh differentiation by OX40L secreted by synovial macrophages may be linked to epigenetic such as histone modifications, which needs to be further explored.

Round 2

Reviewer 2 Report (New Reviewer)

This is now acceptable for publication.

Author Response

Thank you for your review again.  

This manuscript is a resubmission of an earlier submission. The following is a list of the peer review reports and author responses from that submission.

Round 1

Reviewer 1 Report

Lines 83-84: ‘chronic conditions and drug use did not differ significantly between RA and OA group ‘-Define OA group on the first occurrence.

Lines 87- 88: Use the past tense uniformly. 

Lines 112-129: Flow cytometry. Confusing account of flow cytometric analysis. It is stated that ‘flow cytometry was used to analyze sorted cells or single-cell suspension’. However, sell sorting was not described.   

Lines 131-139: The description of PCR analysis is not clear and confusing. Rewrite it. It is not PCR but is an RT-qPCR. What platform was used?

Results: Lines 174-188: are cells sorted? The figure shows two color-analysis on a flow cytometer and not a high-speed cell sorter. Fig. 1H correlation coefficient is poor, r2 is 0.3.

Line 320: What are instrumented mice?

Fig. 6A: Due to the poor quality of the figure, germinal centers cannot be deciphered. Replace the figure.

Lines 354-356: Rewrite the sentence since it is not clear.

Line 365: Substitute the word ‘foundations’ with data to be clear.

Line 378: Not familiar with the term ‘instrumented mice’.

Line 383: Rewrite the sentence for clarity.

General comments: For some reason, the authors state that ‘These suggest that the …….’ Data or results are missing between these and suggest in all sentences throughout the manuscript. It is unclear what the ‘lateral’ corroboration means.  

Reviewer 2 Report

This is a good article. The introduction is very clear and complete. The methods are adequate, the results are significant, and the discussion is appropriate. Therefore, I believe that the article should be accepted for publication in its current form.

Reviewer 3 Report

The manuscript addresses a fascinating subject, as synovial macrophages are a critical factor in the progression to rheumatoid arthritis; in the manuscript titled “Synovial macrophages expression of OX40L is required for follicular helper T cells differentiation in the joint microenvironment” to provide new knowledge in rheumatoid arthritis.

 In the abstract, the authors should clarify RA´s meaning.

It is recommended to the authors to change in 2.7 section “PCR” to “Real-Time PCR” or Reverse Transcriptase-PCR, wherever the case is.

The main feature is vascular hyperplasia: “degree of vascular hyperplasia (0, no vascular hyperplasia; 1, mild hyperplasia; 2 severe hyperplasia; 3 severe hyperplasia)”, what is the difference between degrees 2 and 3? the author should homogenize the terminology to avoid any confusion with the previous feature “synovial cell hyperplasia”.

Lane 153, please change “no lymphocyte” to “no infiltration” (0, no lymphocytes; 1, mild infiltration; 2, moderate infiltration; 3, severe infiltration). 

Lane 173 and 174, the authors mention, “F4/80+ Synovial macrophages (SMs), CD19+ B cells, and CD141+ dendritic cells (DCs) were sorted out from synovial single-cell suspensions. OX40L expression in F4/80+ SMs (Fig. 1A)” but the plot is representing CD11b+ in the X axis and CD192+ in the Y axis, they do not explain whether F4/80+ is equivalent to double marker CD11b+/CD192+. It would be interesting if the authors added a control patient for Fig 1.

In Figure 1B, the cytometry plot does not have the percentage of CD19+ cells, and it looks like de RA synovial tissue has fewer CD19+ cells than OA synovial tissue, while the bar representation for CD19+ and OX40L mRNA is the opposite.

In Figure 1C, the cytometry plot does not consist of CD11c/CD11b+ cells, and there is no d, while the bar representation for CD19+ and OX40L mRNA is the opposite.

Lane 212, the authors mention “three helper T cells (Th1, Tregs, and Tfh)” it should be “three helper T cells types (Th1, Tregs, and Tfh)”.

In Figures 3A to C, please indicate where the effect is located in the images. “The results showed that the severity of arthritis in OX40-/- mice was reduced compared to wild mice (Fig. 3A). The microenvironment of the knee and ankle of OX40-/- mice was improved, as evidenced by reduced inflammatory cell infiltration, reduced bone damage, and reduced synovial hyperplasia (Fig. 3B and C)”. The changes proposed by the authors: inflammatory cell infiltration, reduced bone damage, and reduced synovial hyperplasia, at this amplification, are impossible to see, and the authors should describe the photographs and micrographs to correlate along with the effects in the tissue. For example, Safranin O is a cationic dye that detects variations in articulations. Because Safranin O binds specifically to polyanions to detect the amount of proteoglycan present in cartilages and is used to detect whether glycosaminoglycan depletion in osteoarthritis occurs.

Lane 280, the authors describe “reduced inflammatory cell infiltration (Fig. 4B), and bone damage (Fig. 4C),” but at the amplification of micrographs presented in the manuscript, it is not possible to see the cell infiltration or bone damage. Even all micrographs have not good quality.

The results section is confusing, and the organization could be improved.
